# CLEAR: Context-Aware Learning with End-to-End Mask-Free Inference for Adaptive Video Subtitle Removal

**Qingdong He** [* 1]  **Chaoyi Wang** [* 2]  **Peng Tang** [3]  **Yifan Yang** [4]  **Xiaobin Hu** [5]

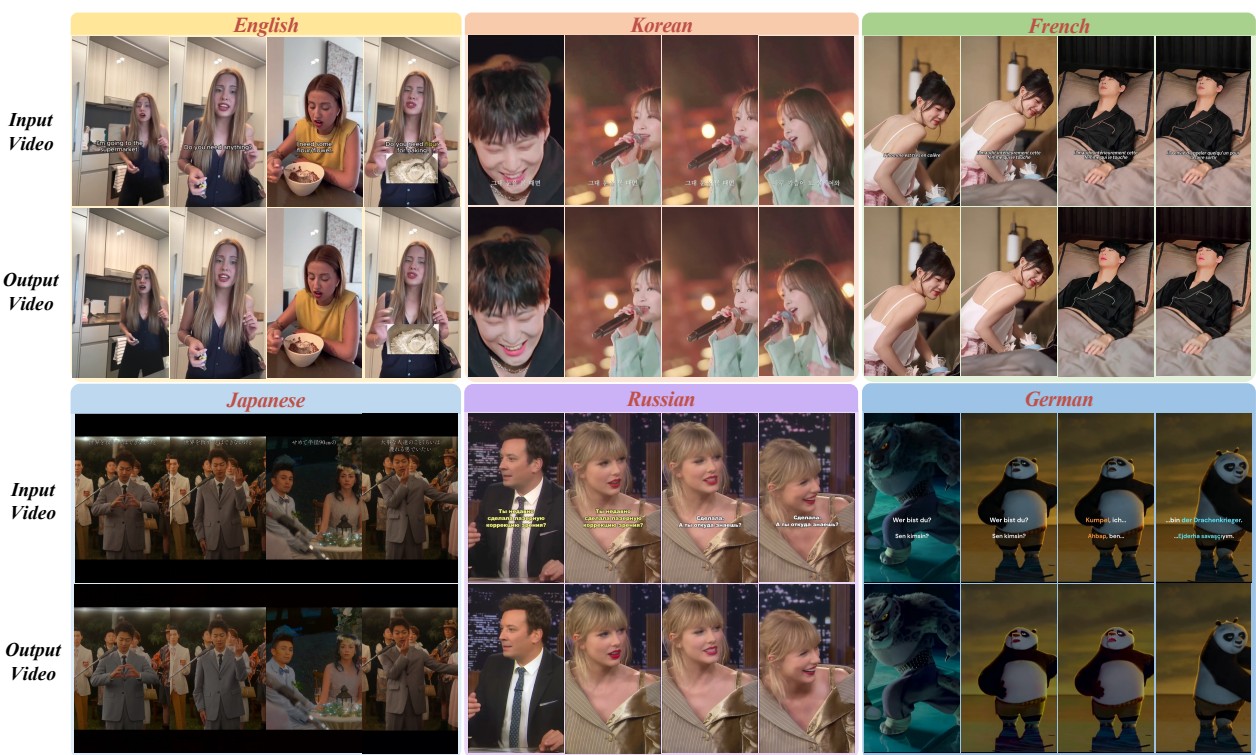

*Figure 1.* **Qualitative visualization of the zero-shot cross-lingual generalization (Zoom in for best view).** CLEAR achieves robust video subtitle removal across English, Korean, French, Japanese, Russian, and German without language-specific training.

## Abstract

Video subtitle removal aims to distinguish text overlays from background content while preserving temporal coherence. Existing diffusion-based methods necessitate explicit mask sequences during both training and inference phases, which restricts their practical deployment. In this paper, we present CLEAR (Context-aware Learning for End-to-end Adaptive Video Subtitle Removal),

*Equal contribution [1]University of Electronic Science and Technology of China [2]University of Chinese Academy of Sciences [3]Technical University of Munich [4]Shanghai Jiao Tong University [5]National University of Singapore. Correspondence to: Xiaobin Hu <ben0xiaobin0hu1@nus.edu.sg>.

*Proceedings of the 43rd International Conference on Machine Learning*, Seoul, South Korea. PMLR 306, 2026. Copyright 2026 by the author(s).

a mask-free framework that achieves truly end-to-end inference through context-aware adaptive learning. Our two-stage design decouples prior extraction from generative refinement: Stage I learns disentangled subtitle representations via self-supervised orthogonality constraints on dual encoders, while Stage II employs LoRA-based adaptation with generation feedback for dynamic context adjustment. Notably, our method only requires **0.77**% of the parameters of the base diffusion model for training. On Chinese subtitle benchmarks, CLEAR outperforms mask-dependent baselines by **+6.77***dB* PSNR and -**74.7**% VFID, while demonstrating superior zero-shot generalization across six languages (English, Korean, French, Japanese, Russian, German)—a performance enabled by our generation-driven feedback mechanism that ensures robust subtitle

removal without ground-truth masks during inference. Code is released.

# 1. Introduction

Video subtitle removal has emerged as a critical task in video content processing with applications in content localization, media re-editing, and multilingual adaptation (Quan et al., 2024; Xu et al., 2019; Yang et al., 2025; Bian et al., 2025). While recent large-scale video diffusion models (Ho et al., 2022; Blattmann et al., 2023; Kong et al., 2024; Wan et al., 2025) provide powerful visual priors for video editing or video inpainting task, video subtitle removal remains challenging due to complex text overlays including semi-transparent fonts, gradient effects, and anti-aliasing that complicate temporally consistent inpainting (Sun et al., 2025).

Current mask-guided diffusion frameworks (Li et al., 2025; Liu & Hui, 2025; Zi et al., 2025) leverage explicit text masks during training and inference. These methods extend image-based techniques (Liu et al., 2020; Peng et al., 2024) with temporal modeling for inter-frame consistency, and disentangled strategies (Zhang et al., 2024) separate text removal from recognition tasks for images.

However, directly applying existing video editing or video inpainting approaches to the task of video subtitle removal face three critical limitations that stem from the temporal and visual properties of subtitles in video. Subtitles exhibit temporal continuity, diverse positions and styles, and complex interactions with camera/object motion, compression artifacts and motion blur; these traits amplify the limitations of current methods: (*L1*) **Training inefficiency.** Existing approaches typically require full model training or fine-tuning of all parameters and rely on dense per-frame mask annotations or specialized segmentation outputs during training (Li et al., 2025; Liu & Hui, 2025; Zi et al., 2025). This cost is substantially higher because consistent, frame-level annotations must be produced or propagated across long sequences. And models must learn to handle temporally varying subtitle appearance (font, size, location) and background dynamics, making large-scale training on unlabelled video infeasible without prohibitive computation or annotation effort. (*L2*) **Inference complexity and fragility.** Many methods depend on explicit mask sequences or external text detection/tracking modules at inference time (Li et al., 2025; Liu & Hui, 2025; Zi et al., 2025). In video applications this introduces additional runtime overhead and engineering dependencies, and renders the pipeline vulnerable to detection failures, false positives, or tracking drift—failure modes that commonly produce flicker, temporal inconsistency, or residual artifacts across frames. (*L3*) **Static prior utilization.** When auxiliary priors (e.g., text heatmaps, segmentation,

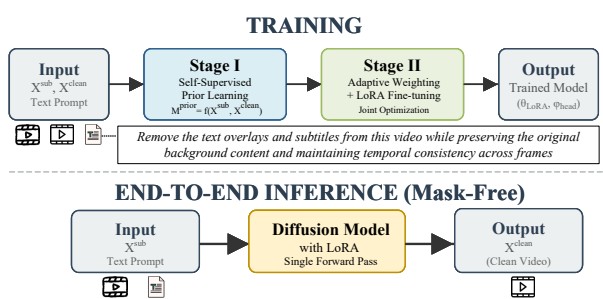

*Figure 2.* **CLEAR framework overview.** Two-stage training enables mask-free inference: Stage I learns self-supervised priors $\mathcal{M}^{prior}$ through disentangled feature learning; Stage II trains LoRA-adapted diffusion with adaptive weighting optimized by distillation, generation feedback, and sparsity losses. Inference requires only subtitled video input.

optical flow) are available, they are often applied uniformly across space and time (Li et al., 2025), ignoring frame- and region-specific reliability. Because subtitle visibility and corruption (occlusion, compression noise, blending with background) vary over time and across regions, a fixed-weight prior can propagate errors or over-smooth content; effective removal therefore requires spatially and temporally adaptive weighting of auxiliary guidance.

Addressing these limitations requires three key capabilities: (*K1*) achieving parameter efficiency and annotation-free learning while preserving pre-trained visual priors and capturing generalizable subtitle features, (*K2*) enabling fully mask-free end-to-end inference without external modules, and (*K3*) adaptively leveraging noisy guidance signals to handle diverse subtitle styles and quality variations.

Therefore, we introduce CLEAR (**C**ontext-aware **L**earning for **E**nd-to-end **A**daptive subtitle **R**emoval), a light-weighted adapter (with an overview in Figure 2), addressing these challenges through three technical innovations. First, self-supervised prior learning (Stage I) extracts occlusion guidance from video pairs using pixel differences as weak supervision—eliminating annotation dependency through orthogonal feature constraints and adversarial purification while learning generalizable subtitle features that transfer across languages and styles. Second, adaptive weighting (Stage II) employs LoRA (Hu et al., 2022) on frozen pre-trained diffusion models with a lightweight occlusion head to dynamically adjust region importance. Training only 0.77% parameters preserves pre-trained priors and enables efficient training. The adaptive mechanism, optimized through structure distillation, generation feedback, and sparsity regularization, robustly handles noisy priors and diverse subtitle characteristics. Third, the trained model achieves end-to-end mask-free inference by internalizing adaptive weighting into LoRA-augmented representations, accepting only video input without external modules. Extensive experiments have shown that our CLEAR, trained on Chinese video subtitle data, not only exhibits superior performance compared to

existing video inpainting methods but also demonstrates excellent zero-shot generalization capabilities to other languages such as English, Japanese, Korean, French, Russian and German. In sum, our contributions are as follows:

- We present CLEAR, a video subtitle removal framework with end-to-end mask-free inference, demonstrating superior performance and strong zero-shot generalization across diverse languages and subtitle styles.
- We propose a novel self-supervised work flow obtaining occlusion priors without mask annotations through disentangled feature learning, enabling annotation-free training and learning generalizable subtitle representations beyond dataset-specific patterns.
- We introduce efficient adaptive weighting via an adapter with unified joint-loss optimization that preserves pre-trained priors while dynamically balancing structure preservation, generation quality, and model stability.

**Conflict of Interest Disclosure.** The authors declare that they have no financial conflicts of interest related to this work.

## 2. Related Work

### 2.1. Text Removal in Images and Videos

Scene text removal methods encompass both image-based and video-based approaches, with increasing focus on end-to-end frameworks and disentangled representation learning. Recent works explore diverse architectures from GANs to vision transformers for enhanced text localization and inpainting quality.

Image-based approaches include EraseNet (Liu et al., 2020), MTRNet (Tursun et al., 2019), and MTRNet++ (Tursun et al., 2020) that separate or unify text localization and inpainting. Liu et al. (Zhang et al., 2024) propose disentangled representation learning for improved controllability across recognition, removal, and editing tasks. ViTEraser (Peng et al., 2024) leverages vision transformers with SegMIM pretraining. To our knowledge, there are currently no methods specifically designed for the task of removing subtitles from videos. Current video methods address temporal consistency through diffusion-based generation: DiffSTR (Pathak et al., 2024) uses masked autoencoder conditioning and MTV-Inpaint (Yang et al., 2025) enables multi-task long video processing.

### 2.2. Diffusion-Based Video Inpainting

Video diffusion models provide powerful generative priors for inpainting tasks, with methods exploring various conditioning strategies, attention mechanisms, and architectural designs. Current approaches incorporate optical flow,

transformer attention, and prior-based guidance for content completion.

Recent large-scale video diffusion models (Ho et al., 2022; Blattmann et al., 2023; Kong et al., 2024) operate in a learned latent space, forming the foundation for modern video synthesis and editing frameworks. ProPainter (Zhou et al., 2023) combines flow propagation with transformers. DiffuEraser (Li et al., 2025) uses DDIM inversion for weak conditioning, while EraserDiT (Liu & Hui, 2025) employs diffusion transformers for temporal consistency. Liu et al. (Liu et al., 2025) reveal through perturbation analysis that high-entropy attention affect video quality while low-entropy maps capture structural information. Advanced strategies include MiniMax-Remover (Zi et al., 2025) with minimax optimization and Zhang et al. (Mao et al., 2026) proposing unified spatially controllable visual effects generation.

Existing video inpainting methods require both mask annotations and full-parameter training, depend on external detection pipelines for inference, and utilize auxiliary priors with uniform weighting. CLEAR addresses these through self-supervised prior learning eliminating annotations, adaptive context-dependent weighting adjusting to prior quality, and fully end-to-end mask-free inference, enabling scalable, robust video subtitle removal.

## 3. Method

### 3.1. Problem Formulation and Overview

CLEAR addresses the mask-free subtitle removal challenge through a carefully designed two-stage framework. Given a video sequence with burned-in subtitles $\mathbf{X}^{sub} = \{x_t^{sub}\}_{t=1}^T \in \mathbb{R}^{T \times H \times W \times 3}$, our goal is to learn a mapping $\mathcal{F} : \mathbf{X}^{sub} \to \mathbf{X}^{clean}$ that recovers the clean video $\mathbf{X}^{clean} = \{x_t^{clean}\}_{t=1}^T$ without requiring pixel-level mask annotations during training or external detection modules during inference. The overall framework of CLEAR is illustrated in Figure 3.

### 3.2. Stage I: Self-Supervised Prior Learning

#### 3.2.1. WEAK SUPERVISION FROM PIXEL DIFFERENCES

Given the video pairs, we first generate pseudo-labels from frame-wise pixel differences:

$$\boldsymbol{\Delta}_t = \|\mathbf{X}_t^{sub} - \mathbf{X}_t^{clean}\|_2 \tag{1}$$

and the binary threshold is defined as:

$$\hat{\mathbf{M}}_t(i,j) = \begin{cases} 1, & \text{if } \boldsymbol{\Delta}_t(i,j) > \mu_t + \sigma_t \\ 0, & \text{otherwise} \end{cases} \tag{2}$$

where $\mu = \mathbb{E}[\boldsymbol{\Delta}]$ and $\sigma = \text{std}[\boldsymbol{\Delta}]$ are computed per-frame. This statistical thresholding identifies high-variance regions

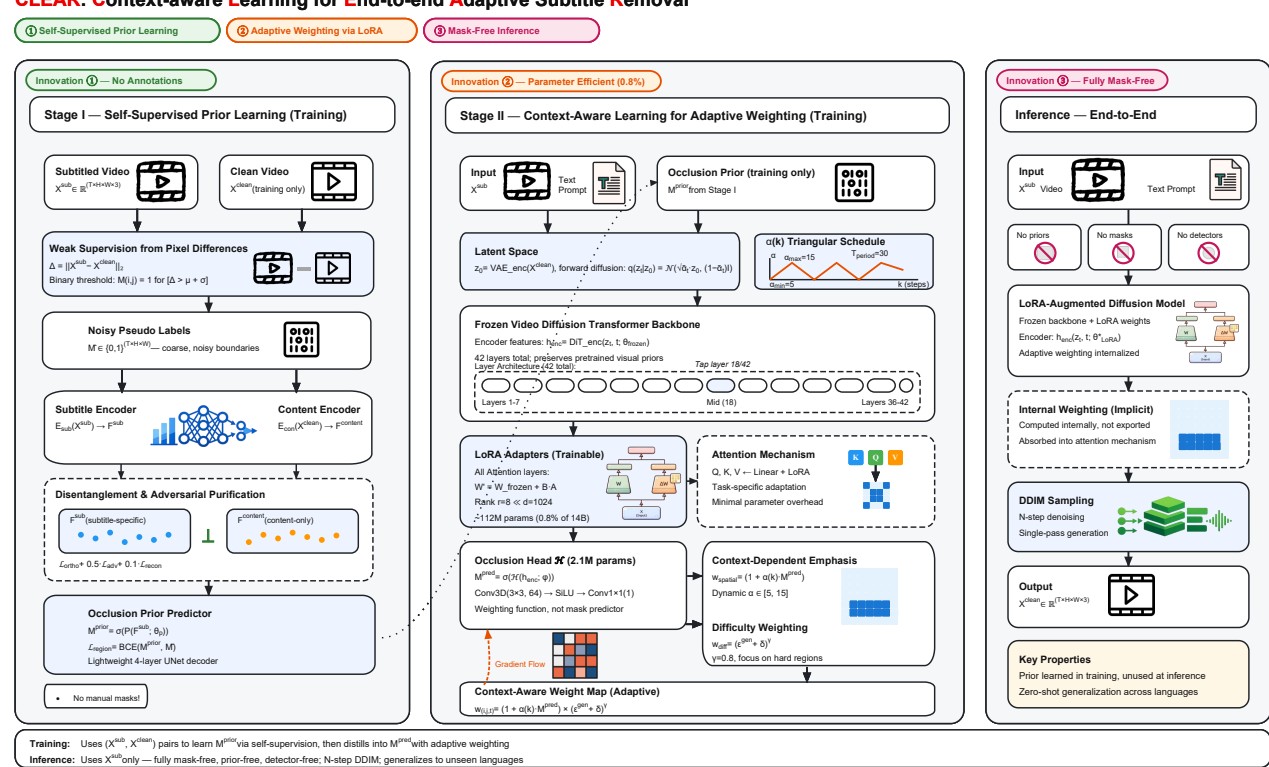

*Figure 3.* **CLEAR Pipeline Details.** Stage I: Dual encoders extract disentangled features with orthogonality and adversarial losses. Stage II: Occlusion head predicts adaptive weights from DiT encoder features, modulating generation through focal weighting $(\epsilon^{gen})^\gamma$ with gradient backflow.

likely corresponding to subtitles, providing binary pseudo-labels $\hat{\mathbf{M}} \in \{0,1\}^{T \times H \times W}$. These labels are intentionally noisy due to lighting changes, semi-transparent subtitles, and motion blur at boundaries. Stage II is designed to robustly handle this noise through adaptive weighting.

### 3.2.2. DISENTANGLED FEATURE LEARNING

We employ dual encoders to separate subtitle-specific and content-specific features at $1/8$ spatial resolution:

$$\mathbf{F}^{sub} = E_{\text{sub}}(\mathbf{X}^{sub}; \theta_s) \in \mathbb{R}^{T \times C \times H' \times W'}$$
$$\mathbf{F}^{content} = E_{\text{content}}(\mathbf{X}^{clean}; \theta_c) \in \mathbb{R}^{T \times C \times H' \times W'} \quad (3)$$

To enforce disentanglement, we minimize feature correlation through orthogonality constraint:

$$\mathcal{L}_{\text{ortho}} = \frac{1}{T \cdot H' \cdot W'} \sum_{t=1}^{T} \sum_{h=1}^{H'} \sum_{w=1}^{W'} \left\langle \mathbf{F}_{t,h,w}^{sub}, \mathbf{F}_{t,h,w}^{content} \right\rangle^2 \quad (4)$$

and adversarial purification via discriminator $D(\cdot; \theta_d)$:

$$\mathcal{L}_{\text{adv}} = -\mathbb{E}[\log D(\mathbf{F}^{sub})] - \mathbb{E}[\log(1 - D(\mathbf{F}^{content}))] \quad (5)$$

Using disentangled features, a lightweight decoder predicts occlusion masks solely from $\mathbf{F}^{sub}$:

$$\mathcal{M}^{prior} = \sigma(P(\mathbf{F}^{sub}; \theta_p)) \quad (6)$$

$$\mathcal{L}_{\text{region}} = \frac{1}{T \cdot H \cdot W} \sum_{t=1}^{T} \sum_{h=1}^{H} \sum_{w=1}^{W} \text{BCE}(\mathcal{M}_{t,h,w}^{prior}, \hat{\mathbf{M}}_{t,h,w}) \quad (7)$$

Content reconstruction verification ensures $\mathbf{F}^{content}$ excludes subtitle information:

$$\mathcal{L}_{\text{recon}} = \frac{1}{T \cdot H \cdot W} \|\text{Dec}(\mathbf{F}^{content}; \theta_{\text{dec}}) - \mathbf{X}^{clean}\|_2^2 \quad (8)$$

The complete Stage I loss balances disentanglement with prediction accuracy:

$$\mathcal{L}_{\text{stage1}} = \mathcal{L}_{\text{ortho}} + 0.5 \cdot \mathcal{L}_{\text{adv}} + \mathcal{L}_{\text{region}} + 0.1 \cdot \mathcal{L}_{\text{recon}} \quad (9)$$

Figure 4 illustrates the mask refinement process. The prior masks $\mathcal{M}^{prior}$ from Stage I capture coarse subtitle regions, which are then refined to $\mathcal{M}^{pred}$ in Stage II through context distillation and generation feedback. Despite imperfections in the pseudo ground-truth masks, our framework effectively learns discriminative subtitle representations for context-aware diffusion guidance.

## 3.3. Stage II: Adaptive Weighting Learning

### 3.3.1. OVERVIEW AND MOTIVATION

Stage II addresses how to robustly utilize noisy priors for high-quality generation, by learning context-dependent

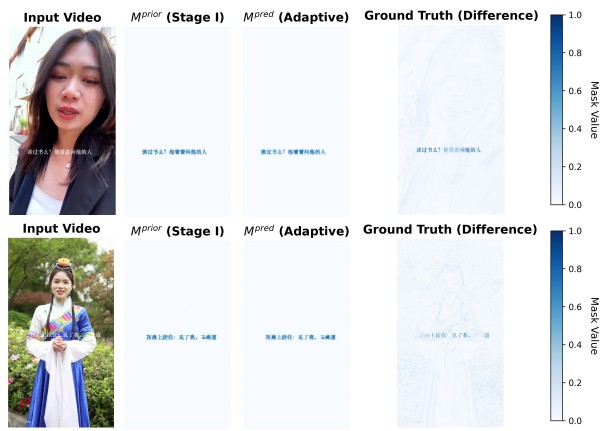

*Figure 4.* Illustration of our progressive mask refinement. Despite noisy pseudo-labels from Stage I ($\mathcal{M}^{prior}$), Stage II learns adaptive weights ($\mathcal{M}^{pred}$) that dynamically adjust based on generation difficulty, enabling robust context-aware modulation, which are internal to training and not predicted during inference.

weighting strategies that adapt based on both prior structure and generation feedback.

### 3.3.2. DIFFUSION MODEL WITH LORA ADAPTATION

We adopt pretrained video diffusion transformer which operates in latent space with its novel Wan-VAE architecture, encoding frames to $\mathbf{z}_0 = \text{VAE}_{\text{enc}}(\mathbf{X}^{clean}) \in \mathbb{R}^{T \times h \times w \times c}$ with spatial compression factor $8\times$ and latent channel dimension $c = 16$.

**Forward Diffusion Process:**

$$q(\mathbf{z}_t|\mathbf{z}_0) = \mathcal{N}(\mathbf{z}_t; \sqrt{\bar{\alpha}_t}\mathbf{z}_0, (1 - \bar{\alpha}_t)\mathbf{I}) \qquad (10)$$

**LoRA Adaptation:** To preserve pretrained visual priors while enabling efficient task-specific adaptation, we apply Low-Rank Adaptation (Hu et al., 2022) to all attention layers in the diffusion transformer:

$$\mathbf{W}' = \mathbf{W}_{\text{frozen}} + \mathbf{BA} \qquad (11)$$

where $\mathbf{A} \in \mathbb{R}^{d \times r}, \mathbf{B} \in \mathbb{R}^{r \times d}$.

### 3.3.3. CONTEXT-DEPENDENT OCCLUSION HEAD

**Architecture Design.** We introduce a lightweight occlusion head $\mathcal{H}$ attached to the diffusion transformer encoder's middle layer:

$$\mathcal{M}^{pred} = \sigma(\mathcal{H}(\mathbf{h}_{\text{enc}}; \phi)) \qquad (12)$$

where $\mathbf{h}_{\text{enc}} = \text{DiT}_{\text{enc}}(\mathbf{z}_t, t; \theta_{\text{LoRA}}) \in \mathbb{R}^{T \times h \times w \times D}$ represents encoder features at diffusion timestep $t$.

**Head Implementation:**

$$\mathcal{H}(\mathbf{h}_{\text{enc}}) = \text{Conv}^1_{1 \times 1}(\text{SiLU}(\text{Conv}^{64}_{3 \times 3}(\mathbf{h}_{\text{enc}}))) \qquad (13)$$

The design allows $\mathcal{M}^{pred}$ to access: (1) low-level texture from noisy latent $\mathbf{z}_t$, (2) high-level semantics from transformer layers, and (3) temporal context from 3D convolutions.

**Conceptual Distinction.** Crucially, $\mathcal{H}$ is not a mask predictor pursuing accuracy—it is a *weighting function* learning to adjust influence based on: (1) local reliability of $\mathcal{M}^{prior}$ via distillation loss, (2) generation difficulty via reconstruction error, and (3) global context via diffusion timestep $t$.

### 3.3.4. ADAPTIVE WEIGHT COMPUTATION

The predicted weights modulate generation through dual adaptation:

$$w_{i,j,t} = \underbrace{(1 + \alpha(k) \cdot \mathcal{M}^{pred}_{i,j,t})}_{\text{spatial emphasis}} \times \underbrace{(\epsilon^{gen}_{i,j,t} + \delta)^{\gamma}}_{\text{difficulty weighting}} \qquad (14)$$

**Spatial Emphasis:** $\alpha(k) \cdot \mathcal{M}^{pred}$ increases attention on predicted subtitle regions, with $\alpha(k)$ controlling emphasis strength.

**Difficulty Weighting:** $(\epsilon^{gen})^{\gamma}$ provides focal-style reweighting based on reconstruction error:

$$\epsilon^{gen}_{i,j,t} = \|\hat{\mathbf{x}}_{i,j,t} - \mathbf{x}^{clean}_{i,j,t}\|^2_2 \qquad (15)$$

where $\hat{\mathbf{x}} = \text{VAE}_{\text{dec}}(\mathbf{z}_0)$ is the generated frame. Hard regions ($\epsilon^{gen}$ large) receive higher weights, while easy regions receive lower weights.

**Dynamic Alpha Scheduling:** To prevent over-reliance on noisy priors, we employ triangular scheduling:

$$\alpha(k) = \alpha_{\min} + (\alpha_{\max} - \alpha_{\min}) \cdot \left| \sin\left(\frac{2\pi k}{T_{\text{period}}}\right) \right| \qquad (16)$$

where $k$ is the training step. This oscillation alternates between emphasizing global features (low $\alpha$) and subtitle regions (high $\alpha$), encouraging exploration and preventing local minima.

### 3.3.5. JOINT OPTIMIZATION WITH CONTEXT-AWARE ADAPTATION

We unify three complementary objectives to train both $\mathcal{H}$ and LoRA weights:

**(1) Context Distillation Loss:**

$$\mathcal{L}_{\text{distill}} = \frac{1}{T \cdot H \cdot W} \sum_{t=1}^{T} \sum_{h=1}^{H} \sum_{w=1}^{W} \text{SmoothL1}(\mathcal{M}^{pred}_{t,h,w}, \mathcal{M}^{prior}_{t,h,w}) \qquad (17)$$

where $\text{SmoothL1}(x) = \begin{cases} 0.5x^2, & |x| < 1 \\ |x| - 0.5, & \text{otherwise} \end{cases}$. This provides soft guidance from Stage I while tolerating small deviations (within 1 unit), giving $\mathcal{M}^{pred}$ flexibility to correct local errors.

**(2) Context-Aware Adaptation Loss:**

$$\mathcal{L}_{\text{gen}} = \mathbb{E}_{t,\mathbf{z}_0,\boldsymbol{\epsilon}} \left[ \frac{1}{T \cdot H \cdot W} \sum_{i=1}^{H} \sum_{j=1}^{W} \sum_{t=1}^{T} w_{i,j,t} \cdot \|\boldsymbol{\epsilon}_\theta(\mathbf{z}_t, t) - \boldsymbol{\epsilon}\|_2^2 \right]$$
(18)

This is the standard diffusion objective weighted by $w_{i,j,t}$. Crucially, $\mathcal{M}^{pred}$ is not detached, allowing gradients to flow:

$$\frac{\partial \mathcal{L}_{\text{gen}}}{\partial \mathcal{M}_{i,j,t}^{pred}} = \alpha(k) \cdot (\epsilon_{i,j,t}^{gen})^{\gamma+1} + \mathcal{O}(\epsilon^{gen}) \qquad (19)$$

where $\mathcal{O}(\epsilon^{gen})$ denotes additional gradient terms from the chain rule that scale with reconstruction error. This enables *context-aware* self-correction: regions with high $\epsilon^{gen}$ receive positive gradients to increase $\mathcal{M}^{pred}$ and allocate more attention, while low-error regions receive negative gradients to decrease $\mathcal{M}^{pred}$ and reduce attention. This feedback loop operates without ground truth masks.

**(3) Context Consistency Loss:**

$$\mathcal{L}_{\text{sparse}} = \underbrace{\frac{1}{T \cdot H \cdot W} \sum_{i=1}^{H} \sum_{j=1}^{W} \sum_{t=1}^{T} \mathcal{M}_{i,j,t}^{pred}}_{\text{L1 sparsity}} \qquad (20)$$

$$+ 0.5 \cdot \underbrace{D_{\text{KL}}(\mathcal{M}^{pred} \| \mathcal{M}^{prior})}_{\text{distribution alignment}} \qquad (21)$$

The L1 term maintains *contextual selectivity*, preventing degeneration to uniform weighting. KL divergence prevents complete deviation from prior structure, ensuring $\mathcal{M}^{pred}$ remains *contextually consistent* with learned subtitle patterns.

**Unified Objective:**

$$\mathcal{L}_{\text{stage2}} = \mathcal{L}_{\text{distill}} + \mathcal{L}_{\text{gen}} + 0.1 \cdot \mathcal{L}_{\text{sparse}} \qquad (22)$$

Equal weights $(1:1)$ for distillation and generation balance structure preservation with performance optimization. Sparsity receives lower weight $(0.1)$ as auxiliary regularization.

### 3.4. End-to-End Mask-Free Inference

During inference, the trained model operates in a fully mask-free manner:

---
**Algorithm 1** Mask-Free Inference

---
**Input:** $\mathbf{X}^{sub}$ (subtitled video only)
**Initialization:** $\mathbf{z}_T \sim \mathcal{N}(\mathbf{0}, \mathbf{I}), \quad \mathbf{z}_0^{\text{sub}} = \text{VAE}_{\text{enc}}(\mathbf{X}^{sub})$
**Denoising:**
1: **for** t = T, . . . , 1 **do**
2:    $\mathbf{h}_{\text{enc}} = \text{DiT}_{\text{enc}}(\mathbf{z}_t, t; \theta_{\text{LoRA}}^*)$
3:    $\mathcal{M}^{pred} = \sigma(\mathcal{H}(\mathbf{h}_{\text{enc}}; \phi^*))$   (internal, not output)
4:    $\mathbf{z}_{t-1} = \text{DDIM}_{\text{step}}(\mathbf{z}_t, t; \theta_{\text{LoRA}}^*, \phi^*)$
5: **end for**
**Output:** $\mathbf{X}^{clean} = \text{VAE}_{\text{dec}}(\mathbf{z}_0)$

---

**Key Properties:** (1) **No Stage I Dependency:** $\mathcal{M}^{prior}$ is not required—$\mathcal{H}$ implicitly captures subtitle patterns from encoder features alone; (2) **No External Modules:** No text detection, no segmentation models, no auxiliary networks; (3) **Internalized Weighting:** Although $\mathcal{M}^{pred}$ is computed internally, it never appears in the output—the adaptive strategy is absorbed into LoRA-augmented attention maps; (4) **Single-Pass Generation:** One forward pass directly maps $\mathbf{x}_{\text{sub}} \rightarrow \hat{\mathbf{x}}_{\text{clean}}$ in a fully end-to-end manner.

## 4. Experiments

### 4.1. Experimental Setup

**Baseline Methods.** We compare CLEAR against three state-of-the-art video inpainting methods: ProPainter (Zhou et al., 2023), MiniMax-Remover (Zi et al., 2025), and DiffuEraser (Li et al., 2025). All baselines use official implementations with default hyperparameters. For comparison, we provide identical binary masks generated via the same thresholding procedure used in CLEAR's Stage I training: $\hat{\mathbf{M}}_t(i,j) = 1$ if $\|\mathbf{X}_t^{sub} - \mathbf{X}_t^{clean}\|_2 > \mu_t + \sigma_t$, where $\mu_t$ and $\sigma_t$ denote mean and standard deviation of pixel-wise differences.

**Evaluation Metrics.** We employ a comprehensive evaluation protocol spanning four categories: **(1) Reconstruction Quality:** PSNR and SSIM (Wang et al., 2004) measure pixel-level fidelity, while LPIPS (Zhang et al., 2018) captures perceptual similarity via deep features. **(2) Perceptual Quality:** DISTS (Ding et al., 2020) evaluates structure and texture preservation, and VFID (Unterthiner et al., 2018) assesses video-level distribution alignment. **(3) Temporal Consistency:** TWE (Temporal Warping Error) (Lai et al., 2018) detects frame-to-frame flickering via optical flow, and TC (Temporal Coherence) measures adjacent frame differences. **(4) Motion Smoothness:** Flow Mean and Flow Variance quantify optical flow stability across frames.

### 4.2. Implementation Details.

All experiments are conducted on a compute cluster equipped with 8 GPUs. We use PyTorch 2.0 with mixed-precision training (bfloat16) to accelerate computation and reduce memory footprint. The base video diffusion model is Wan2.1-Fun-V1.1-1.3B (Wan et al., 2025), a lightweight variant with 1.3 billion parameters optimized for controllable video generation. The model operates in latent space with Wan-VAE architecture, using a spatial compression factor of $8\times$ and latent channel dimension $c = 16$. Due to the absence of publicly available benchmark datasets for video subtitle removal, we collect over 160,000 pairs of data for model training, including subtitled videos with Chinese fonts of various styles and sizes. From this dataset, we constructed a benchmark of 400 random samples to facilitate fair comparison across models.

**Stage I: Self-Supervised Prior Learning.** In Stage I, we train a disentangled representation learning framework to

extract occlusion guidance from video pairs without manual annotations. We employ dual ResNet-50 (He et al., 2016) encoders pretrained on ImageNet: a subtitle encoder $E_{\text{sub}}$ processing subtitle frames and a content encoder $E_{\text{content}}$ processing clean frames. Both encoders extract features at $1/8$ spatial resolution, producing $\mathbf{F}^{sub}, \mathbf{F}^{content} \in \mathbb{R}^{T \times C \times H' \times W'}$.

To enforce feature disentanglement, we apply orthogonality constraints minimizing inner products between corresponding feature locations, and adversarial purification via a 3-layer CNN discriminator that distinguishes subtitle features from content features. A lightweight 4-layer UNet decoder predicts binary subtitle masks $\mathcal{M}^{prior} \in [0,1]^{T \times H \times W}$ solely from $\mathbf{F}^{sub}$, while a reconstruction decoder verifies that $\mathbf{F}^{content}$ excludes subtitle information by reconstructing clean frames.

We train on 500 video pairs comprising real-world footage from internet sources (400 pairs) and synthetic green-screen recordings (100 pairs). Each video is uniformly sampled to extract 20 frames, yielding 10,000 training frame pairs. Pseudo-labels are generated via statistical thresholding on pixel-wise differences: $\hat{\mathbf{M}}_t(i,j) = 1$ if $\Delta_t(i,j) > \mu_t + \sigma_t$, where $\Delta_t = \|\mathbf{X}_t^{sub} - \mathbf{X}_t^{clean}\|_2$. We use a resolution-adaptive strategy that resizes videos to $1280 \times 720$ for landscape orientation and $720 \times 1280$ for portrait, preserving aspect ratios. The effective batch size is 192 (6 per GPU $\times$ 8 GPUs $\times$ 4 gradient accumulation steps).

The Stage I objective combines four losses with empirically tuned weights:

$$\mathcal{L}_{\text{stage1}} = \mathcal{L}_{\text{ortho}} + 0.5 \cdot \mathcal{L}_{\text{adv}} + \mathcal{L}_{\text{region}} + 0.1 \cdot \mathcal{L}_{\text{recon}} \quad (23)$$

where $\mathcal{L}_{\text{ortho}}$ enforces orthogonality between $\mathbf{F}^{sub}$ and $\mathbf{F}^{content}$, $\mathcal{L}_{\text{adv}}$ provides adversarial purification, $\mathcal{L}_{\text{region}}$ applies binary cross-entropy on mask prediction, and $\mathcal{L}_{\text{recon}}$ ensures clean frame reconstruction from content features. We optimize with AdamW (Loshchilov & Hutter, 2019) using a learning rate of $2 \times 10^{-5}$ with 500 warmup steps and cosine decay, training for 1 epoch ($\sim$70 minutes on our hardware).

**Stage II: Adaptive Guidance Learning.** In Stage II, we freeze the Stage I encoders to serve as the prior generator and train lightweight LoRA (Hu et al., 2022) adapters injected into the video diffusion model. We set the LoRA rank to 64 and apply low-rank decomposition $\mathbf{W}' = \mathbf{W}_{\text{frozen}} + \mathbf{BA}$ to all attention components (q, k, v, o) and feed-forward network layers (ffn.0, ffn.2) in the transformer blocks, where $\mathbf{A} \in \mathbb{R}^{d \times r}$, $\mathbf{B} \in \mathbb{R}^{r \times d}$ with $r = 64 \ll d = 1024$.

We introduce a context-dependent occlusion head $\mathcal{H}$ with only 2.1M parameters attached to the diffusion transformer's middle encoder layer. The head consists of two convolutional layers: $\mathcal{H}(\mathbf{h}_{\text{enc}}) = \text{Conv}_{1 \times 1}^1(\text{SiLU}(\text{Conv}_{3 \times 3}^{64}(\mathbf{h}_{\text{enc}})))$,

where $\mathbf{h}_{\text{enc}}$ represents encoder features at diffusion timestep $t$ with embedding dimension $D = 1024$. The predicted weights $\mathcal{M}^{pred} = \sigma(\mathcal{H}(\mathbf{h}_{\text{enc}}))$ modulate generation through adaptive focal weighting:

$$w_{i,j,t} = (1 + \alpha(k) \cdot \mathcal{M}_{i,j,t}^{pred}) \times (\epsilon_{i,j,t}^{gen} + \delta)^{\gamma} \quad (24)$$

where $\alpha(k)$ provides dynamic emphasis scheduling via triangular oscillation between $\alpha_{\min} = 5$ and $\alpha_{\max} = 15$ over period $T_{\text{period}} = 40$ iterations, $\epsilon_{i,j,t}^{gen} = \|\hat{\mathbf{x}}_{i,j,t} - \mathbf{x}_{i,j,t}^{clean}\|_2^2$ measures reconstruction error, and we set $\gamma = 0.8$, $\delta = 10^{-6}$.

Training uses 500 carefully curated video samples, each with 81 consecutive frames to capture longer temporal dependencies. The effective batch size is 8 (1 per GPU $\times$ 8 GPUs $\times$ 1 gradient accumulation). We employ uniform timestep sampling to ensure balanced coverage of the diffusion process and apply random blackout augmentation to the predicted masks for robustness.

The Stage II objective unifies three complementary losses:

$$\mathcal{L}_{\text{stage2}} = \mathcal{L}_{\text{distill}} + \mathcal{L}_{\text{gen}} + 0.1 \cdot \mathcal{L}_{\text{sparse}} \quad (25)$$

where $\mathcal{L}_{\text{distill}}$ applies smooth L1 loss between $\mathcal{M}^{pred}$ and $\mathcal{M}^{prior}$ for soft structural guidance while tolerating local deviations, $\mathcal{L}_{\text{gen}}$ is the standard diffusion objective weighted by $w_{i,j,t}$ with gradients flowing through $\mathcal{M}^{pred}$ to enable self-correction based on reconstruction error, and $\mathcal{L}_{\text{sparse}}$ combines L1 sparsity regularization with KL divergence $D_{\text{KL}}(\mathcal{M}^{pred} \| \mathcal{M}^{prior})$ to prevent degeneration and maintain distribution alignment. We optimize with AdamW using a learning rate of $1 \times 10^{-4}$, gradient clipping at norm 1.0, and enable gradient checkpointing for memory efficiency. Models are trained for 1 epoch with checkpoints saved every 25 steps, taking approximately 1 day for full convergence on our 8-GPU setup.

**Input Prompt Design.** Following standard video inpainting practices (Kong et al., 2024; Wan et al., 2025), we adopt a concise text prompt for Stage II inference:

> *"Remove the text overlays and subtitles from this video while preserving the original background content and maintaining temporal consistency across frames."*

Unlike mask-dependent baselines (Li et al., 2025; Liu & Hui, 2025), CLEAR performs fully mask-free inference without encoding any spatial mask information in the prompt. The adaptive weighting mechanism is implicitly learned during training and absorbed into LoRA-augmented attention, where $\mathcal{M}^{pred}$ is generated internally from encoder features without relying on $\mathcal{M}^{prior}$ at inference time.

### 4.3. Main Results

**Quantitative Comparison.** Table 1 presents quantitative comparisons with state-of-the-art methods. CLEAR

*Table 1.* Quantitative comparison with state-of-the-art methods on Chinese subtitle test set. CLEAR uses default configuration (rank=64, steps=5, cfg=1.0, lora_scale=1.0).

| Method | Reconstruction | | | Perceptual | | Temporal | | Flow | | Time |
|---|---|---|---|---|---|---|---|---|---|---|
| | PSNR↑ | SSIM↑ | LPIPS↓ | DISTS↓ | VFID↓ | TWE↓ | TC↓ | Mean↓ | Var↓ | s/frame↓ |
| ProPainter (Zhou et al., 2023) | 17.24 | 0.658 | 0.3291 | 0.245 | 98.46 | 1.286 | 1.160 | 0.952 | 0.885 | 2.36 |
| Minimax-Remover (Zi et al., 2025) | 20.03 | 0.773 | 0.166 | 0.119 | 95.39 | 4.222 | 3.016 | 0.841 | 0.415 | 4.90 |
| DiffuEraser (Li et al., 2025) | 17.85 | 0.672 | 0.458 | 0.228 | 72.51 | 1.523 | 1.174 | 0.840 | 0.630 | 3.47 |
| **CLEAR (Ours)** | **26.80** | **0.894** | **0.101** | **0.075** | **20.37** | **1.227** | **1.049** | **0.209** | **0.029** | 4.86 |

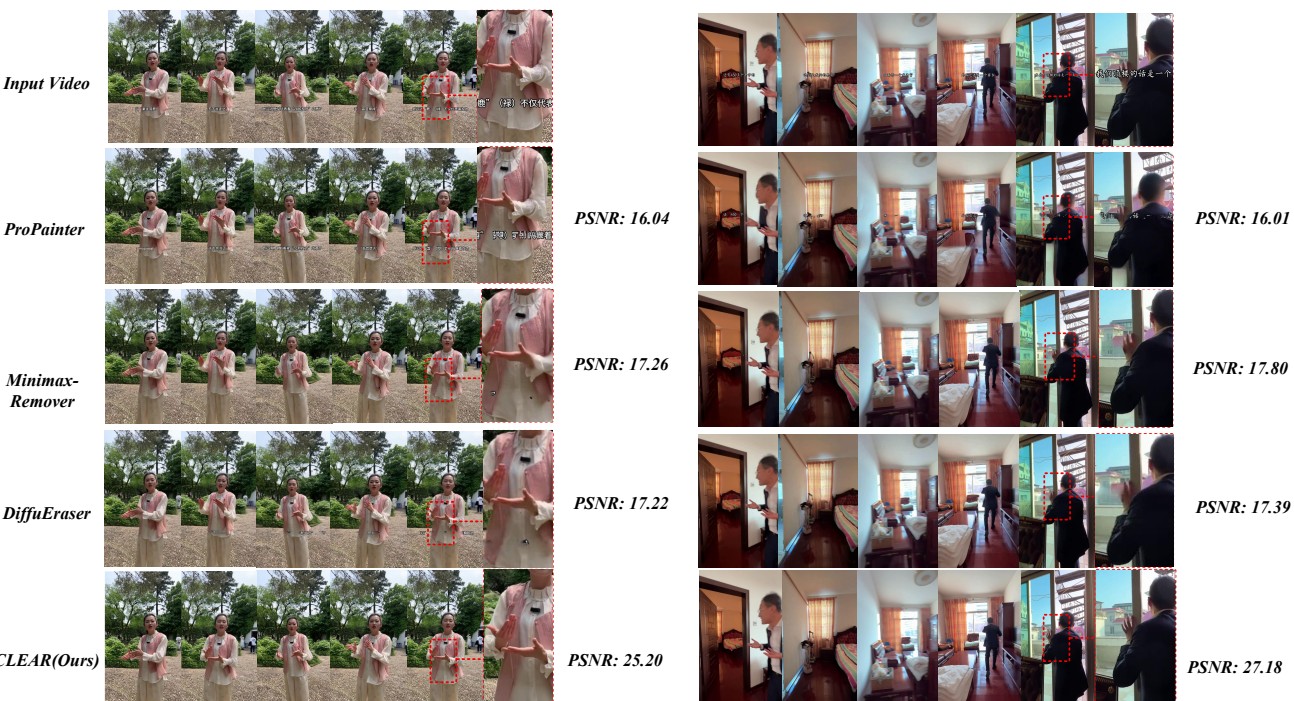

*Figure 5.* **Qualitative comparison with baseline methods.** Unlike ProPainter, Minimax-Remover, and DiffuEraser that require explicit masks during inference, CLEAR achieves mask-free end-to-end removal with clean subtitle elimination and fine-grained detail preservation.

achieves substantial improvements across all metrics: **+6.77 dB PSNR** over the best baseline (Minimax-Remover), **-74.7% VFID** reduction indicating superior perceptual quality, and **-93.0% Flow Variance** demonstrating exceptional temporal stability. Notably, CLEAR maintains competitive inference speed (4.86 s/frame) despite requiring no explicit mask sequences during inference, unlike all baselines which depend on externally provided binary masks. The consistent superiority across reconstruction, perceptual, and temporal metrics validates our context-aware adaptation mechanism.

**Qualitative Comparison.** As shown in Figure 5 and Figure A1, CLEAR outperforms baseline methods in visual performance across diverse subtitle scenarios. Baseline methods suffer from noticeable flaws: they either leave residual subtitle traces or text outlines, introduce background blurring, or cause over-smoothing that loses fine-grained details, with some exhibiting inconsistent brightness between inpainted regions and surrounding contexts. In con-

trast, CLEAR achieves complete subtitle removal without residues while faithfully preserving the original background texture, color, and structural details. It maintains excellent temporal coherence, ensuring smooth frame-to-frame transitions without flickering or texture mismatches—advantages rooted in its mask-free end-to-end design. The visual results confirm that CLEAR strikes an optimal balance between removal completeness, background preservation, and temporal stability, aligning with its superior quantitative metrics.

**Cross-Lingual Generalization.** Figure 1, Figure A2, Figure A3 and Figure A4 demonstrate zero-shot performance on six languages (English, Korean, French, Japanese, Russian, German) unseen during training. CLEAR consistently removes subtitles across diverse scripts and text layouts without language-specific fine-tuning, validating the language-agnostic nature of our self-supervised prior learning. This generalization stems from learning abstract occlusion patterns rather than character-specific features.

*Table 2.* Ablation study on CLEAR's modular design. Each row progressively adds one component.

| Configuration | Reconstruction | | | Perceptual | | Temporal | | Flow | |
|---|---|---|---|---|---|---|---|---|---|
| | PSNR↑ | SSIM↑ | LPIPS↓ | DISTS↓ | VFID↓ | TWE↓ | TC↓ | Mean↓ | Var↓ |
| Baseline (LoRA-only) | 21.62 | 0.855 | 0.131 | 0.088 | 34.74 | 1.320 | 1.137 | 0.217 | 0.032 |
| + M1: Stage I Prior with Focal Weighting | 23.11 | 0.868 | 0.130 | 0.091 | 38.21 | 1.303 | 1.123 | 0.216 | 0.031 |
| + M2: Context Distillation | 24.72 | 0.890 | 0.110 | 0.083 | 31.73 | 1.279 | 1.102 | 0.212 | 0.030 |
| + M3: Context-Aware Adaptation | 25.09 | 0.891 | 0.109 | 0.083 | 31.56 | 1.257 | 1.082 | 0.211 | 0.029 |
| + M4: Context Consistency (CLEAR) | 26.80 | 0.894 | 0.101 | 0.075 | 20.37 | 1.227 | 1.049 | 0.209 | 0.029 |

*Table 3.* Results of key inference hyperparameters on performance.

| Configuration | Reconstruction | | | Perceptual | | Temporal | | Flow | | Time |
|---|---|---|---|---|---|---|---|---|---|---|
| | PSNR↑ | SSIM↑ | LPIPS↓ | DISTS↓ | VFID↓ | TWE↓ | TC↓ | Mean↓ | Var↓ | s/frame↓ |
| *Denoising Steps (lora_scale=1.0, cfg=1.0)* | | | | | | | | | | |
| steps=5 | 26.80 | 0.894 | 0.101 | 0.075 | 20.37 | 1.227 | 1.049 | 0.209 | 0.029 | 4.86 |
| steps=10 | 29.43 | 0.884 | 0.101 | 0.081 | 35.70 | 1.358 | 1.243 | 0.215 | 0.030 | 9.92 |
| *CFG Scale (lora_scale=1.0, steps=5)* | | | | | | | | | | |
| cfg=0.8 | 25.24 | 0.899 | 0.109 | 0.083 | 30.05 | 1.257 | 1.078 | 0.210 | 0.028 | 4.86 |
| cfg=1.2 | 29.65 | 0.874 | 0.107 | 0.087 | 40.71 | 1.284 | 1.181 | 0.206 | 0.026 | |
| *LoRA Scale (cfg=1.0, steps=5)* | | | | | | | | | | |
| lora_scale=0.5 | 25.17 | 0.841 | 0.184 | 0.115 | 63.02 | 1.829 | 1.879 | 0.375 | 0.114 | 4.86 |
| lora_scale=1.5 | 27.94 | 0.869 | 0.116 | 0.087 | 42.16 | 1.409 | 1.311 | 0.220 | 0.031 | |

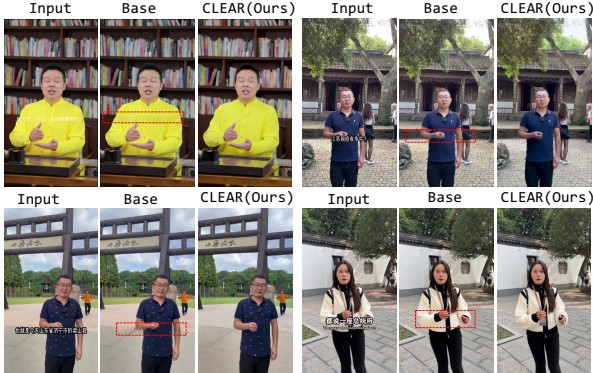

*Figure 6.* Visual comparison of output quality between the LoRA-only baseline and CLEAR (Ours). The baseline struggles with residual subtitles and background blurring, while our two-stage framework achieves clean subtitle removal and faithful preservation of structural details.

### 4.4. Ablation Study

**Ablation of Components.** Table 2 demonstrates the progressive contribution of each module. The baseline LoRA-only adaptation achieves 21.62 dB PSNR, with each subsequent component providing incremental gains: Stage I prior with focal weighting (M1) improves PSNR by +1.49 dB through coarse localization; context distillation (M2) adds +1.61 dB by refining mask predictions; context-aware adaptation (M3) contributes +0.37 dB via generation feedback; and context consistency regularization (M4) yields a final +1.71 dB boost alongside a dramatic 35.5% VFID reduction. The cumulative 5.18 dB PSNR improvement from baseline to full CLEAR demonstrates the synergy of our two-stage design. Figure 6 visually validates these gains, contrasting the LoRA-only baseline and CLEAR. The base-

line leaves subtitle residues and background blurring, while CLEAR achieves complete subtitle removal and preserves fine-grained details—aligning with Table 2's quantitative results and confirming the effectiveness of our proposed modules.

**Hyperparameter Analysis.** Table 3 analyzes key inference parameters. Increasing denoising steps from 5 to 10 improves PSNR (+2.63 dB) but degrades temporal consistency (TWE +10.7%) and doubles inference time, confirming our default choice of 5 steps. CFG scale exhibits a quality-diversity trade-off: cfg=1.0 balances fidelity and perceptual quality, while higher values introduce temporal artifacts. LoRA scale significantly impacts performance, with lora_scale=1.0 providing optimal balance—lower values (0.5) yield incomplete removal (LPIPS +82%), while higher values (1.5) cause over-smoothing.

## 5. Conclusion

We introduce CLEAR, a mask-free video subtitle removal framework that achieves end-to-end inference through context-aware adaptive learning. By decoupling prior extraction from generative refinement, our two-stage design requires only 0.77% trainable parameters while outperforming mask-dependent baselines by a large margin. The self-supervised prior learning enables zero-shot cross-lingual generalization, and the generation feedback mechanism provides dynamic context adaptation without ground-truth masks. Future work includes extending to other video occlusion removal tasks and exploring real-time inference optimization.

## Impact Statement

We plan to make the dataset and associated code publicly available for research. Nonetheless, we acknowledge the potential for misuse, particularly by those aiming to generate misinformation using our methodology. We will release our code under an open-source license with explicit stipulations to mitigate this risk.

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

## Appendix

### A. Two-Stage Training Pipeline

**Stage I (Self-Supervised Prior Learning):** Extract coarse occlusion guidance from paired videos using pixel differences as weak supervision. This stage learns where subtitles likely appear without requiring precise boundaries:

$$\mathcal{M}^{prior} = f_{\text{prior}}(\mathbf{X}^{sub}, \mathbf{X}^{clean}; \Theta_1) \tag{A1}$$

where $\mathcal{M}^{prior} \in [0,1]^{T \times H \times W}$ provides spatial-temporal occlusion guidance, and $\Theta_1 = \{\theta_s, \theta_c, \theta_p, \theta_d\}$ are learnable parameters.

**Stage II (Adaptive Weighting Learning):** Train a diffusion model with LoRA adaptation and a lightweight occlusion head that learns to adaptively utilize the noisy priors:

$$\mathcal{M}^{adapt} = f_{\text{weight}}(\mathbf{z}_t, t, \mathcal{M}^{prior}; \Theta_2), \tag{A2}$$

$$\mathbf{X}^{clean} = f_{\text{diffusion}}(\mathbf{X}^{sub}, \mathcal{M}^{adapt}; \Theta_2) \tag{A3}$$

where $\Theta_2 = \{\theta_{\text{LoRA}}, \phi_{\text{head}}\}$ consists of low-rank adaptation matrices and occlusion head parameters. Crucially, $\mathcal{M}^{adapt}$ is predicted internally and learns to adjust its influence based on generation quality.

**Inference (End-to-End Mask-Free):** The trained model internalizes the adaptive weighting mechanism:

$$\mathbf{X}^{clean} = f_{\text{diffusion}}(\mathbf{X}^{sub}; \Theta_2^*) \tag{A4}$$

No external modules or Stage I priors are required. The occlusion head implicitly captures subtitle patterns from input features alone.

### B. More Qualitative Results

To further demonstrate the performance of CLEAR compared to SOTA methods, we present additional qualitative results in Figure A1, Figure A2, Figure A3 and Figure A4 . It is evident that CLEAR consistently outperforms other SOTA approaches, producing more coherent and visually plausible edits. These results highlight CLEAR's ability to perform full subtitle removal with zero residues, maintaining the background's texture, color, and structural integrity.

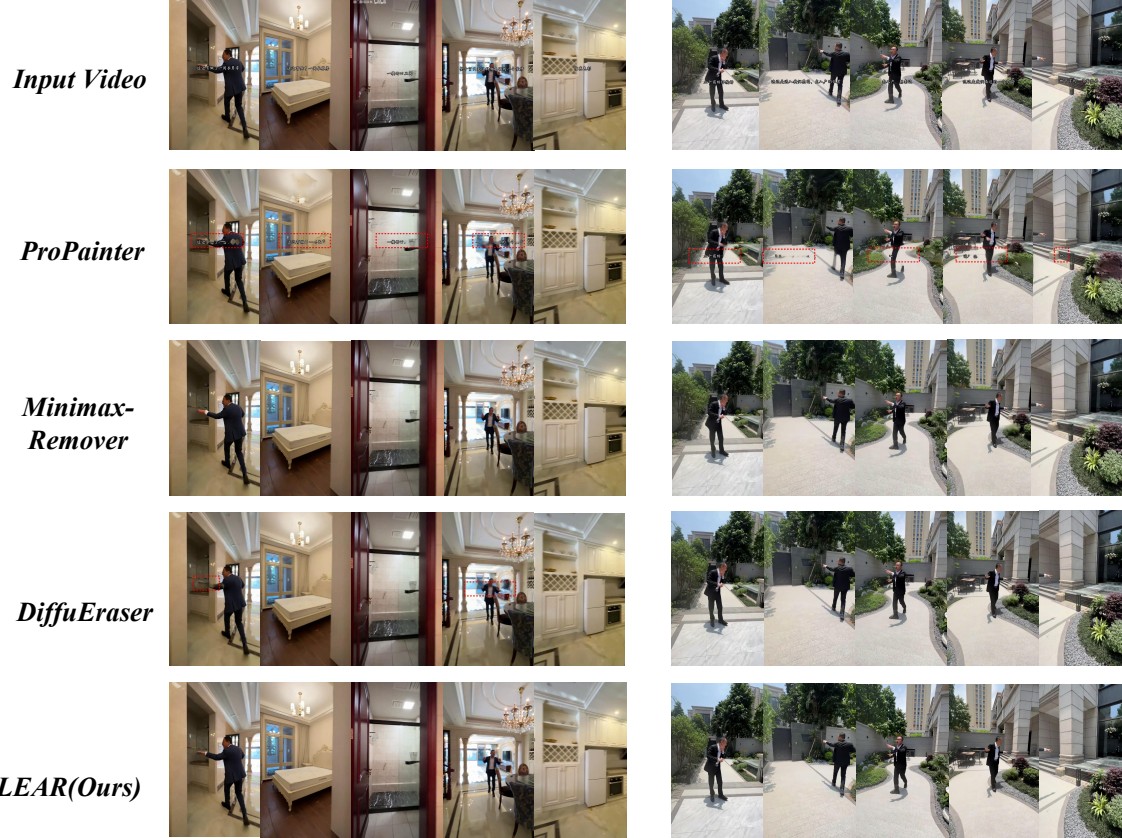

*Figure A1.* **Qualitative comparison with baseline methods.** Unlike ProPainter, Minimax-Remover, and DiffuEraser that require explicit masks during inference, CLEAR achieves mask-free end-to-end removal with clean subtitle elimination and fine-grained detail preservation.

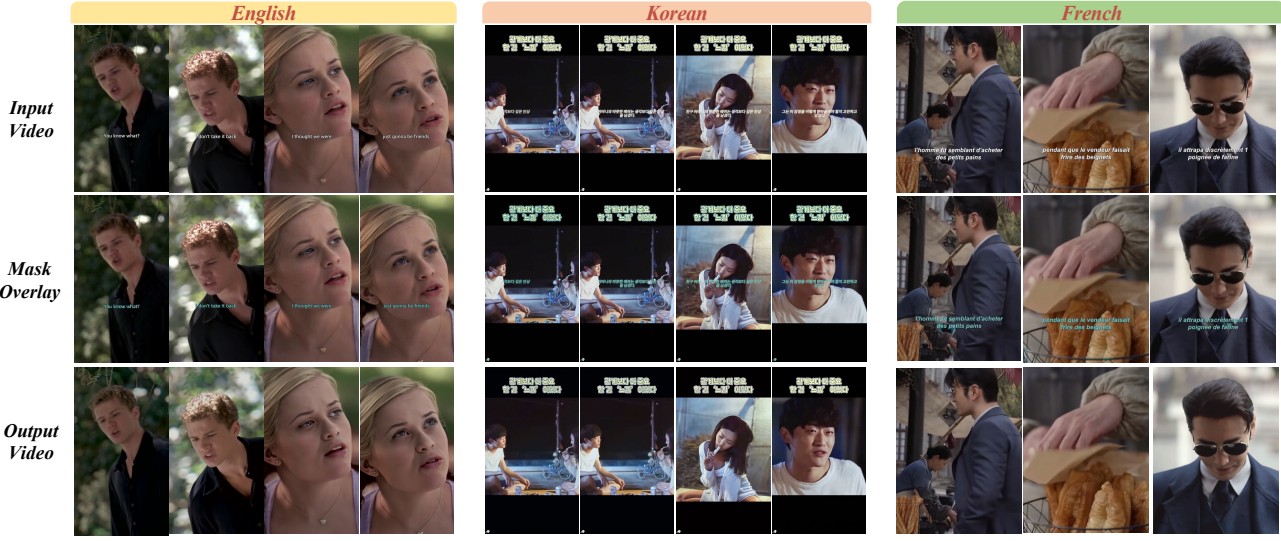

*Figure A2.* **Zero-shot cross-lingual generalization.** CLEAR achieves robust subtitle removal across English, Korean, French, Japanese, Russian, and Arabic without language-specific training. The mask overlays visualize the learned context-aware guidance ability during Stage II training, demonstrating accurate subtitle localization; notably, these masks are **not** predicted during inference.

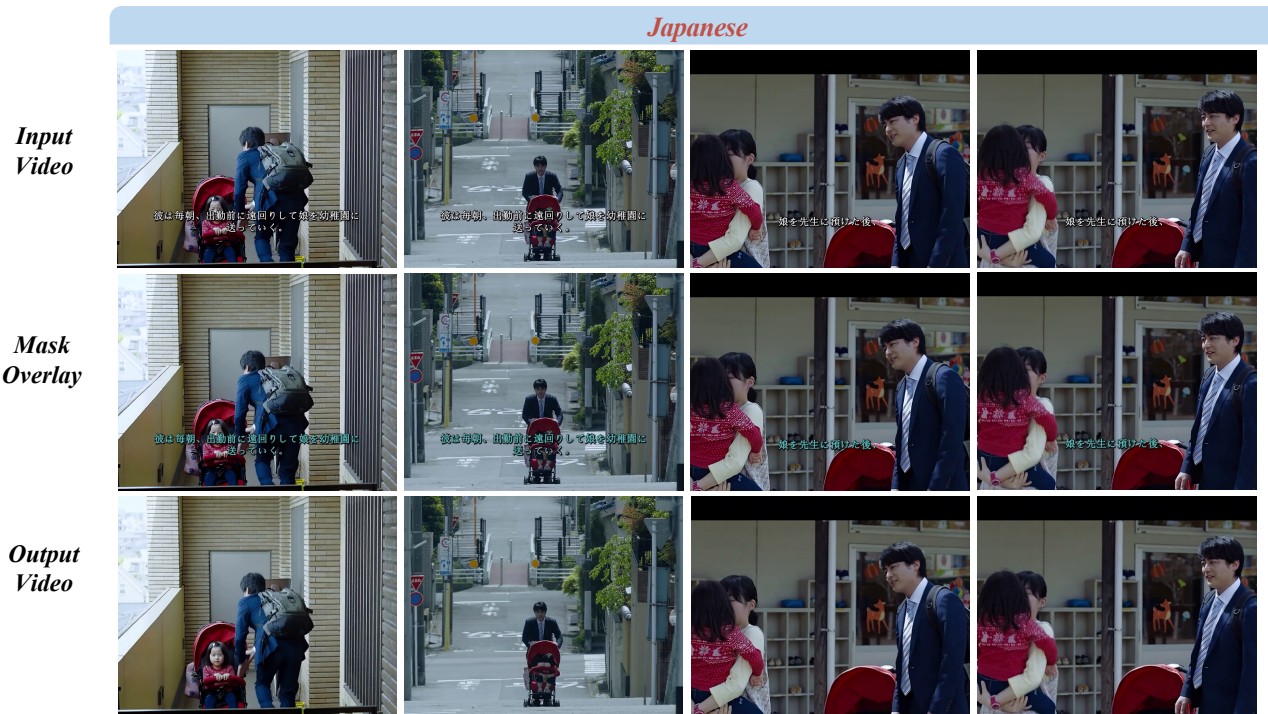

*Figure A3.* **Zero-shot cross-lingual generalization.** CLEAR achieves robust subtitle removal across English, Korean, French, Japanese, Russian, and Arabic without language-specific training. The mask overlays visualize the learned context-aware guidance ability during Stage II training, demonstrating accurate subtitle localization; notably, these masks are **not** predicted during inference.

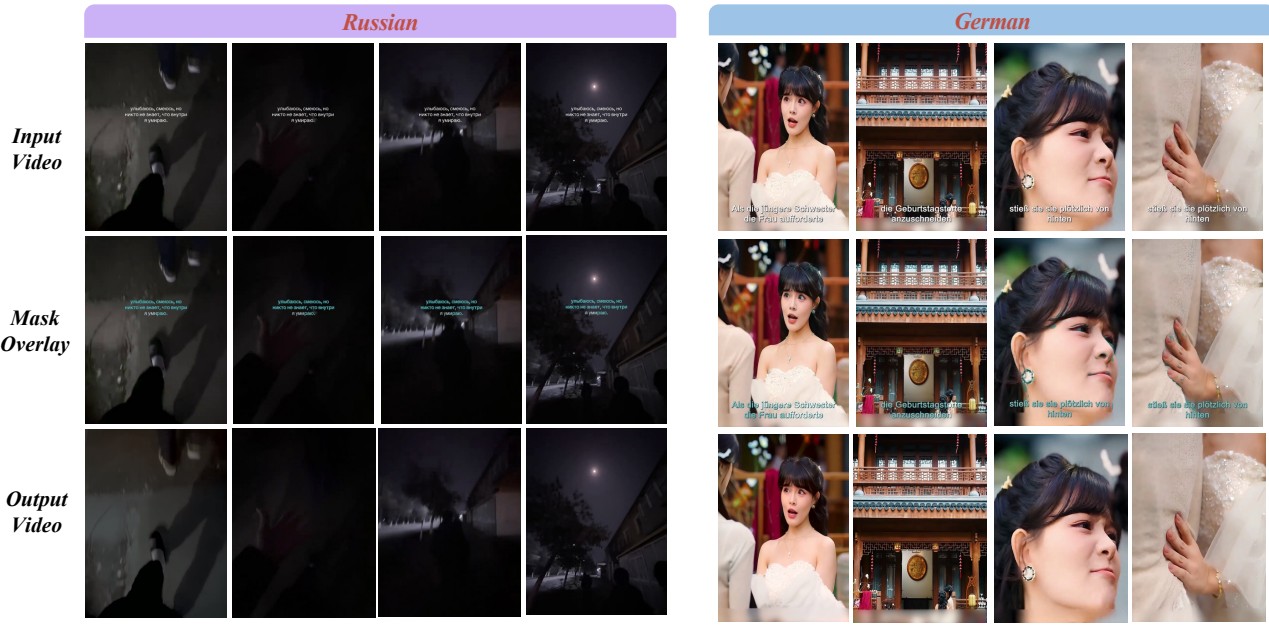

*Figure A4.* **Zero-shot cross-lingual generalization.** CLEAR achieves robust subtitle removal across English, Korean, French, Japanese, Russian, and Arabic without language-specific training. The mask overlays visualize the learned context-aware guidance ability during Stage II training, demonstrating accurate subtitle localization; notably, these masks are **not** predicted during inference.

