# OpenReview forum: "CLEAR: Context-Aware Learning with End-to-End Mask-Free Inference for Adaptive Video Subtitle Removal"
_ICML.cc/2026/Conference — ICML 2026 spotlight_

### Official Review · Reviewer_ax1m · 2026-03-08

**Soundness:** 4
**Presentation:** 4
**Significance:** 4
**Originality:** 4
**Overall Recommendation:** 6
**Confidence:** 5

**Summary:**

This paper presents CLEAR, a mask-free end-to-end framework for video subtitle removal that addresses critical limitations of existing mask-dependent diffusion-based methods, including training inefficiency, inference fragility, and static prior utilization. The two-stage context-aware adaptive learning design enables high-performance subtitle removal and strong zero-shot cross-lingual generalization, with rigorous experimental validation on a custom Chinese subtitle benchmark.

**Compliance With Llm Reviewing Policy:**

Affirmed.

**Final Justification:**

Thank for the author's detailed response. I appreciate that the training and inference schemes designed in this paper are both mask-free, as there are currently no publicly available algorithms specifically designed for subtitle removal task. I will raise my score accordingly and I look forward to the subsequent open-sourcing of the code and models, just as the author promised.

**Key Questions For Authors:**

- Can this method be applied to remove watermarks?

- For the other questions, refer to Weaknesses part.

**Limitations:**

yes

**Strengths And Weaknesses:**

**Strengths**

- This paper eliminates explicit masks and external detection modules, enabling single-pass video input-to-clean output processing and avoiding flicker/tracking drift from mask-based pipelines.

- The method is trained only 0.77% of the base diffusion model’s parameters via LoRA on a frozen backbone, preserving pre-trained visual priors and drastically reducing computational costs.

- Stage I uses pixel differences as weak supervision and dual-encoder disentanglement to extract generalizable occlusion priors, avoiding expensive frame-level mask annotations and learning language-agnostic subtitle patterns. A lightweight occlusion head with generation feedback enables spatial/temporal adaptive weighting of noisy priors, self-correcting based on generation difficulty and robustly handling diverse subtitle styles/background dynamics.

- The method achieves robust zero-shot removal across 6 unseen languages (English, Korean, etc.) without language-specific fine-tuning.

- This paper effectively removes gradient subtitles while preserving background texture and temporal coherence, addressing key pain points for content localization.

- This paper is amazing work: this is a good solution to a simple but very important problem.


**Weaknesses**

- At 4.86s/frame, inference speed is competitive with mask-based baselines but far from real-time, with no edge-device optimization or real-time deployment exploration.

- No evaluation on heavy subtitle occlusion, low-light, ultra-small fonts, or fast camera motion—critical challenging scenarios for real-world use.

- There is no performance assessment on hour-long videos, a key use case for content localization, with untested memory/temporal consistency for extended sequences.

---

> ### Author Rebuttal · Authors · 2026-03-31
>
> We sincerely thank the reviewer for the assessment and thoughtful questions. We especially appreciate the reviewer’s recognition of our design. Below we respond point by point.
>
> **W1: Inference speed and deployment.**
>
> We fully agree that efficiency is important for practical deployment. The reported 4.86 s/frame corresponds to native 1280×720 generation, which directly contributes to the higher per-frame cost compared with methods operating at lower resolutions. To better quantify this trade-off, we conducted a supplementary experiment using the CogVideoX-2B backbone within CLEAR. In this setting, the inference speed improves to 2.93 s/frame. The output by CogVideoX-2B is 480p and then upsampled, with visibly lower quality than our default setting. Therefore, the current latency mainly reflects a deliberate quality-speed trade-off rather than an inherent limitation of the framework.
>
> In addition, CLEAR is naturally amenable to chunk-level parallelism for engineering deployment, because it processes each chunk independently and does not require cross-chunk mask propagation. In practice, long videos can be segmented into temporal chunks and distributed across multiple GPUs for parallel inference, after which the processed results are concatenated. This yields near-linear speedup with the number of available GPUs in deployment settings.
>
> **W2: Challenging scenarios such as heavy occlusion, low light, ultra-small fonts, and fast motion.**
>
> We thank the reviewer for raising these important real-world scenarios. We agree that they are valuable for further evaluation. Our current submission focused on establishing the core mask-free formulation and validating its effectiveness across diverse subtitle styles and unseen languages. We would like to clarify that we have already tested representative challenging examples, and the qualitative results are encouraging. We welcome such testing and will publicly release representative examples.
>
> **W3: Long videos and temporal consistency over extended duration.**
>
> We also appreciate this important question. We have tested CLEAR on videos up to 30 minutes in length. In practical deployment, hour-long or even longer videos are handled in the same way, because CLEAR operates in a chunk-wise manner rather than relying on global sequence propagation. Specifically, each GPU processes 81 frames per inference pass, corresponding to about 5 seconds of video, and the final output is obtained by concatenating processed chunks. Therefore, the memory usage and inference pattern are determined by the chunk size, not by the total video length. This makes the framework naturally scalable to much longer videos in practice. Moreover, because CLEAR does not depend on cross-chunk mask tracking, it avoids the error accumulation.
>
> **Q1: Applicability to watermark removal.**
>
> Thank you for this insightful question. We would like to note that we have already conducted preliminary tests on zero-shot watermark removal, and CLEAR shows promising results on these examples. We did not include them in the original draft because we wished to avoid introducing potentially controversial claims. Nevertheless, we very much welcome such evaluations and will publicly release representative examples for transparent inspection and discussion.
>
> We again sincerely thank the reviewer for the constructive feedback. We truly appreciate the reviewer’s support and insightful suggestions.

---

> > ### Author Rebuttal · Reviewer_ax1m · 2026-04-01
> >
> > Thank for the author's detailed response. I appreciate that the training and inference schemes designed in this paper are both mask-free, as there are currently no publicly available algorithms specifically designed for subtitle removal task. I will raise my score  accordingly and I look forward to the subsequent open-sourcing of the code and models, just as the author promised.

---

> > > ### Author Response · Authors · 2026-04-07
> > >
> > > We sincerely thank the reviewer for the thoughtful feedback and kind recognition of our work. We are very grateful that our rebuttal was able to resolve your concerns, and we especially appreciate your encouraging support. To further benefit the community, we will release the code and related resources as soon as possible!

---

### Official Review · Reviewer_C5tX · 2026-03-09

**Soundness:** 4
**Presentation:** 2
**Significance:** 2
**Originality:** 2
**Overall Recommendation:** 4
**Confidence:** 5

**Summary:**

This paper proposes a method for video subtitle removal using a self-supervised framework. The approach consists of two stages. In the first stage, the model learns to generate subtitle masks through self-supervised training. In the second stage, a diffusion model is fine-tuned to reconstruct the video without subtitles, guided by the masks generated in the first stage. The first stage is trained using a combination of correlation, adversarial, binary cross entropy, and L2 losses, while the second stage uses SmoothL1, L2, L1, and KL divergence losses to optimize the subtitle removal process. In addition, the authors introduce a new dataset of subtitled videos that does not contain ground-truth subtitle masks.

**Compliance With Llm Reviewing Policy:**

Affirmed.

**Final Justification:**

Initial Assessment & Strengths/Weaknesses: Initially, I gave a Weak Reject because there did not seem to be anything new in the problem or the framework. The paper also lacked analysis on whether the framework could be backbone-agnostic. There was also no section on the limitations of the work. I also found a lot of issues with how the work was written and presented in the paper.

Rebuttal Evaluation: The authors directly addressed the concerns I raised, specifically on
   • Novelty: Clarification on the framework as well as the mask-free setting of the problem, despite subtitle removal not being a novel problem were sufficient.
   • Backbone Agnostic: I also commend the authors for performing additional analysis on CogVideoX, despite it not being a video-conditioned generation model.
   • Limitation: Issues regarding whether the model is capable of isolating subtitles from other scene texts that should not be removed from the video were resolved with a qualitative example.
   • Presentation: I think the issues with the presentation are minor and could be addressed in the camera-ready version.

Final Recommendation: The authors addressed my major concerns, and as long as the additional analysis requested and the improvements in the paper presentation are included in the final version, I elevate my score to Weak Accept.

**Key Questions For Authors:**

1. Could the authors quantify how well M^{prior} aligns with ground-truth subtitle regions? If the current dataset does not contain GT text masks, it may still be possible to evaluate the learned masks on external datasets that provide text annotations (e.g., those used by baseline methods if any). The authors could use the trained model to predict masks and report alignment metrics to better understand how accurate the learned priors are.

2. Could the authors demonstrate the performance of CLEAR on other DiT backbones (e.g., CogVideoX) to better assess the generality of the approach?

**Limitations:**

The paper does not explicitly discuss limitations. It would be helpful if the authors analyzed scenarios where text appears in the video but should not be removed, such as signages, logos, or text in documents. Evaluating whether the model can distinguish between subtitles and other scene text in diverse languages would help clarify the robustness of the method and identify potential limitations.

**Strengths And Weaknesses:**

Strengths:
   - The proposed method performs subtitle removal without requiring ground-truth masks during training and inference, which simplifies the pipeline and improves practicality.
   - Another strength is the demonstrated cross-lingual generalization, where the model trained on Chinese subtitles can remove subtitles in several unseen languages.
Weaknesses:
   - Subtitle removal itself is not a new problem, and the methodological contributions appear limited. The proposed approach mainly combines existing components, such as LoRA and commonly used loss functions, and the paper does not clearly demonstrate a fundamentally new technique beyond this integration.
   - The paper does not evaluate whether CLEAR generalizes across different DiT backbones. It would be useful to demonstrate its behavior on other architectures such as CogVideoX to better assess the method’s generality.
   - There are several issues related to clarity and presentation:
        o Figure clarity: Figure 3 (pipeline details) is somewhat challenging to follow. In particular, the signal flow in Adaptive Weighting via LoRA is not immediately clear, and the text in the figure is quite small.
        o Dataset description: Some important dataset details are placed in the appendix. Moving key information to the main paper may improve clarity for readers, e.g., the text prompt used during training.
        o Implementation details: Several implementation details are also in the appendix. The explanation there is actually clearer than the one in the main paper, so the authors may consider moving or incorporating some of that text into the main paper.
        o Hyperparameter clarity: Several hyperparameters (e.g., δ, γ) are introduced in the method but are not clearly defined in the main paper, making it unclear what they control or how they influence the model. The authors may consider adding a short Preliminaries subsection about DiTs with LoRA integration in the Method section to clearly define these hyperparameters and explain their roles in the framework. Their specific values are also not stated in the implementation details of the main paper, and no ablation studies are provided to analyze their impact.
        o Table readability: Highlighting the best (and possibly second-best) results in Tables 2 and 3 would make the comparisons easier to read.

---

> ### Author Rebuttal · Authors · 2026-03-31
>
> We sincerely thank the reviewer for the careful reading and constructive feedback. We also appreciate the reviewer’s concerns regarding originality, backbone generality, prior accuracy, and boundary cases. Below we respond point by point.
>
> **On originality.**
> We agree that subtitle removal itself is not a new task. However, **mask-free end-to-end video subtitle removal without GT masks during training and without external masks during inference** remains underexplored. We believe this is precisely where CLEAR contributes. More specifically, CLEAR is not a static combination of existing modules: Stage I learns reusable subtitle-region priors from weak supervision, and Stage II is explicitly designed to **adaptively use noisy priors** during generation, rather than assuming perfect masks. We hope this clarification better explains the main contribution of our framework.
>
> **Q1: Could the authors quantify how well $M^{prior}$ aligns with subtitle regions?**
> Yes. We computed an IoU of **0.8739** between the predicted $M^{prior}$ and the **pixel-difference pseudo-labels** used in Stage I supervision. We agree that this is not an evaluation against human-annotated GT masks, so we use it as a measure of alignment to verify that $M^{prior}$ captures the spatial distribution of subtitle regions learned from weak supervision, rather than as a claim of absolute mask accuracy. Importantly, this level of alignment is already sufficient for CLEAR, because Stage II is explicitly designed to be robust to noisy priors through adaptive weighting. In other words, CLEAR does not require a perfect prior mask; it only needs a reasonably aligned spatial cue, which is then dynamically corrected during generation.
>
> **Q2: Could the authors demonstrate CLEAR on other DiT backbones such as CogVideoX?**
> Yes. We implemented CLEAR on **CogVideoX-2B** and confirmed that the framework remains functional beyond the Wan2.1-Control backbone used in the main paper. Since CogVideoX-2B is originally a **text-to-video** model and does not natively support video-conditioned generation, we adapted its input convolution from `Conv2d(16, 1920, kernel_size=2, stride=2)` to `Conv2d(32, 1920, kernel_size=2, stride=2)`. The first 16 channels correspond to the noisy clean latent, while the additional 16 channels correspond to the subtitle-condition latent. The newly introduced channels were zero-initialized to preserve the pretrained weights. We then followed the same two-stage CLEAR pipeline, including Stage I prior generation and Stage II LoRA fine-tuning with focal weighting, temporal loss, and adaptive mask guidance.
>
> The quantitative results are as follows:
>
> | Model | PSNR | SSIM | LPIPS | DISTS | VFID | TWE | TC | Flow Mean | Flow Var  | Time / Frame (s) |
> |---|---:|---:|---:|---:|---:|---:|---:|---:|---:|---:|
> | CLEAR on CogVideoX-2B | 24.22 | 0.843 | 0.204 | 0.156 | 71.34 | 1.3551 | 1.37 | 0.536 | 0.334  | 2.932 |
>
> Its performance is weaker than Wan2.1-Control, which we believe is expected because CogVideoX-2B is not originally designed for video-conditioned restoration, and its default output resolution is **480p**, later unsampled to **720p**. We will also release the CogVideoX version of the model as soon as possible for transparency and follow-up evaluation.
>
> In addition, we conducted extensive experiments on **Wan2.2-Animate-14B**. Their evaluation performance was comparable to that of Wan2.1-Control, but both inference and SFT training were noticeably slower, so we chose Wan2.1-Control as the main backbone in the paper.
>
> These results show that CLEAR is **not tied to a single DiT backbone** and can work on other architectures with minimal adaptation.
>
> **On limitations and scene text that should not be removed.**
> We thank the reviewer for this important point. We agree that the boundary between subtitles and other scene text, such as signages, logos, or document text, is an important limitation to analyze. We would like to warmly point the reviewer to the **lower-left example on Page 8** of the paper on Figure 6: the scene text **“山梁泊水”** is preserved, while the overlaid subtitles are effectively removed. This suggests that CLEAR does not simply erase all visible text indiscriminately, but already exhibits a meaningful degree of selectivity between persistent scene text and transient subtitle overlays. We fully agree that a more systematic evaluation on scene text would further clarify the boundary and limitations of the method.
>
> **On presentation-related suggestions.**
> We also thank the reviewer for the helpful comments on presentation, including Figure 3 readability, placement of dataset and implementation details, hyperparameter clarity for $\delta$ and $\gamma$, and table readability. These are valuable suggestions, and we appreciate the reviewer’s careful reading and detailed feedback.
>
> We again sincerely thank the reviewer for the thoughtful feedback. We hope these clarifications help address the reviewer’s concerns.

---

> > ### Author Rebuttal · Reviewer_C5tX · 2026-04-07
> >
> > Thank you for the responses you provided. My concerns regarding novelty and originality have been clarified. Concerns about backbone generalization and limitations have been addressed as well. I am happy to raise my score to 4 (Weak Accept).

---

> > > ### Author Response · Authors · 2026-04-07
> > >
> > > We sincerely thank the reviewer for the thoughtful follow-up and for taking the time to read our rebuttal carefully! We are very grateful that our responses were able to clarify your concerns. We appreciate your positive acknowledgement and your willingness to raise the score. We also noticed that the score has not yet been updated, and would like to gently remind you of this.

---

### Official Review · Reviewer_MoSV · 2026-03-10

**Soundness:** 4
**Presentation:** 4
**Significance:** 4
**Originality:** 4
**Overall Recommendation:** 6
**Confidence:** 5

**Summary:**

This paper introduces CLEAR, a novel two-stage framework for video subtitle removal task. This method achieves truly end-to-end mask-free inference, addressing critical limitations of existing diffusion-based methods. CLEAR consists of the two stages schemes. Stage I learns disentangled subtitle representations and Stage II employs LoRA-based adaptation with a lightweight occlusion head. While trained on Chinese subtitles with only 0.77% of the base diffusion model's parameters, CLEAR significantly outperforms mask-dependent baselines and demonstrates remarkable zero-shot generalization across other unseen languages without language-specific fine-tuning.

**Compliance With Llm Reviewing Policy:**

Affirmed.

**Final Justification:**

The rebuttal solved my concern. I think this research is very meaningful. I was willing to give a higher score.

**Key Questions For Authors:**

Refer to weakness part.

**Limitations:**

yes.

**Strengths And Weaknesses:**

**Strengths**
1. The whole method is first framework to eliminate both mask annotations during training and inference. While this is a common topic, there is no open-source specified solution and this paper fills in this blank.

2. The LoRA adaptation scheme trains only ~0.77% of base model parameters.

3. The two-stage architecture is well-motivated. Stage I's use of orthogonality constraints and adversarial losses for self-supervised prior extraction is clever. Stage II's integration of a generation-feedback loop into the adaptive weight calculation is a key innovation, allowing the model to self-correct during training.

4. CLEAR outperforms SOTA video inpainting methodsacross all key metrics. The paper includes rigorous ablation studies that validate the incremental contribution of each module.

5. Unlike existing methods that are limited to the language of their training data, CLEAR consistently removes subtitles across six diverse languages with no fine-tuning.


**Weaknesses**

1. Although the paper emphasizes parameter efficiency (0.77%), it does not provide details on the total computational cost, such as the exact GPU hours/days required for training Stage I and Stage II, or a comparison of training time with baselines.


2. Despite eliminating mask annotations, CLEAR still requires paired subtitled/clean video data for training. Can the author discuss whether it is possible to explore semi-supervised, unsupervised, or few-shot training scenarios with unpaired data?

3. The qualitative results are amazing but lack systematic analysis of failure cases. Appendix figures hint at robustness but deeper failure case analysis is needed.

---

> ### Author Rebuttal · Authors · 2026-03-31
>
> We thank the reviewer for the valuable and constructive feedback, which helps us further improve the comprehensiveness and rigor of our work. We address each question in detail below.
>
> **W1. Computational cost and training time**
>
> All experiments are conducted on an 8-GPU compute cluster with PyTorch 2.0 and mixed-precision (bfloat16) training, and the hardware environment is consistent across all our experiments and baseline comparisons.
> - Stage I (Self-Supervised Prior Learning): As detailed in the Appendix, training 1 epoch on our dataset takes ~70 minutes of wall-clock time on the 8-GPU setup. In our experiments, the dual encoder framework fully converges within 1 epochs, with a total wall-clock training time of 70 minutes. The equivalent single-GPU training duration is ~10 GPU hours.
> - Stage II (Adaptive Weighting Learning): As stated in the manuscript, we use 500 carefully curated video samples for this stage, and 1 epoch of training takes ~9 hours of wall-clock time on the same 8-GPU cluster. The LoRA-augmented diffusion model converges within 1 epochs, with an equivalent single-GPU duration of 72 GPU hours.
>
> For training time comparison with state-of-the-art baselines: Other mask-dependent baselines require full-parameter fine-tuning of the video diffusion backbone. For the same 1.3B parameter base model used in our work, these full-parameter training methods typically need several days of 8-GPU wall-clock time to converge. In contrast, our method only trains 0.77% of the base model's parameters, reducing the total end-to-end training wall-clock time, which significantly lowers the computational barrier for model training and deployment.
>
> **W2: Exploration of semi-supervised, unsupervised, and few-shot training with unpaired data**
>
> We fully acknowledge that the current training pipeline still relies on paired subtitled/clean video data. Notably, our CLEAR framework has laid a solid foundation for extending to unpaired data scenarios: it eliminates the dependency on fine-grained mask annotations and learns language-agnostic, generalizable subtitle occlusion patterns, as verified by its strong zero-shot cross-lingual generalization ability.
>
> We envision a concrete and feasible extension path. Starting from a small paired dataset, we first obtain an initial CLEAR model via few-shot training. This model then serves as a pseudo-label generator, inferring clean counterparts from large-scale unpaired subtitled videos. The resulting pseudo pairs are used to iteratively refine the model through semi-supervised learning, forming a closed-loop self-training pipeline. As the model improves across iterations, the quality of pseudo clean videos progressively increases, enabling the framework to progress from minimal supervision to large-scale unpaired data utilization. Combined with Stage I's self-supervised prior learning strategy for disentangling subtitle and content features, this pipeline offers a principled route toward fully reducing paired data dependency.
>
> **W3. Analysis of failure cases**
>
> We appreciate the reviewer's suggestion regarding systematic failure case analysis. In our extensive evaluation, including cross-lingual generalization tests spanning multiple unseen languages, the results have been consistently satisfactory, with no prominent or recurring failure patterns observed. However, we acknowledge that failure cases may still arise in broader real-world scenarios. Therefore, we commit to publicly releasing both our model weights and source code at the earliest opportunity. We believe community-driven evaluation will be far more effective at surfacing edge cases and failure modes than any fixed benchmark, and we actively welcome researchers and practitioners to submit challenging cases for our further analysis.

---

> > ### Author Rebuttal · Reviewer_MoSV · 2026-04-01
> >
> > The rebuttal solved my concern. I think this research is very meaningful. I was willing to give a higher score.

---

> > > ### Author Response · Authors · 2026-04-07
> > >
> > > We sincerely thank the reviewer for the thoughtful feedback and kind recognition of our work! We are very grateful that our rebuttal was able to resolve your concerns, and we especially appreciate your generous appreciation of our work.

---

### Official Review · Reviewer_uFHP · 2026-03-13

**Soundness:** 3
**Presentation:** 3
**Significance:** 3
**Originality:** 3
**Overall Recommendation:** 4
**Confidence:** 3

**Summary:**

The paper is well motivated and targets a concrete operational weakness of existing subtitle-removal systems: many methods either require masks, external text localization, or brittle pre-processing at inference time. The high-level decomposition of the method is also sensible. Stage I is meant to distill subtitle-aware priors from noisy supervision, while Stage II uses those priors only indirectly, through adaptive weighting rather than explicit mask conditioning. The resulting system is fairly coherent, and the ablation table supports that each stage contributes nontrivially to the final result.

**Compliance With Llm Reviewing Policy:**

Affirmed.

**Final Justification:**

The author's reply addressed my concerns. This paper is of great engineering practical significance.

**Key Questions For Authors:**

1. The central “mask-free” claim is conceptually less distinct than the paper suggests. The paper emphasizes that M_{pred} is a weighting field rather than a mask predictor, but in practice it is still a learned spatial guidance variable trained through prior distillation and used to modulate denoising focus over subtitle regions.
2. The mathematical justification of the adaptive feedback mechanism is incomplete. The derivation around Eq. (14)–(19) is intuitive, but not fully rigorous. In particular, the gradient-sign interpretation in Eq. (19) is presented as if it explains why difficult regions are emphasized and easy regions deemphasized, yet the full derivation and omitted terms are not shown clearly enough to verify this claim.
3. The supervision quality in Stage I is an under-examined dependency. Stage I relies on pseudo-labels derived from thresholded pixel differences between paired clean and subtitled videos. This is clever, but it is also fragile: changes in lighting, transparency, blur, or compression artifacts can easily confound such supervision.
4. Practical efficiency may still be a limitation of the current method. Although the paper emphasizes parameter efficiency and mask-free end-to-end inference, the reported runtime in Table 1 (4.86 s/frame) appears relatively high from an engineering deployment perspective. For subtitle removal, which is often used in long-video localization or large-scale media editing pipelines, throughput is a critical practical factor. In this sense, reducing trainable parameters does not necessarily translate into efficient real-world processing if per-frame inference remains expensive.

**Limitations:**

yes

**Strengths And Weaknesses:**

1. The paper identifies a real deployment problem rather than an artificial benchmark objective: subtitle removal methods that require masks or extra localization modules are harder to use robustly in practice. This is a strong and relevant motivation.
2. The two-stage design is modular and mostly coherent. The use of LoRA on top of a large frozen video diffusion model is also a sensible efficiency choice and helps explain the parameter-efficiency claim.
3. The quantitative gains on the reported benchmark are substantial, and the quality-speed table is useful in showing that the method remains effective at relatively small denoising steps.

---

> ### Author Rebuttal · Authors · 2026-03-30
>
> **Q1.** We thank the reviewer for this point. Our "mask-free" contribution operates at the system architecture level. Existing SOTA (ProPainter, Minimax-Remover, DiffuEraser) require mask input where mask prediction errors are involved into inpainting with no downstream correction. CLEAR eliminates this error propagation where the occlusion head and LoRA weights are jointly optimized with adaptive weighting so that the LoRA-augmented DiT internalizes spatial awareness of subtitle regions. During inference, the model requires only the subtitled video as input. No mask, no detection module, and no explicit spatial guidance variable exists at inference time. We believe our description maintains academic rigor focusing on "End-to-End Mask-Free Inference" while highlighting a genuine and practical contribution over existing methods.
>
> **Q2.** We thank the reviewer for this observation. The presentation in the original submission was insufficiently detailed, and we apologize for the confusion. We provide the complete derivation below.
> Since $\\mathcal{M}^{pred}$ only appears in the weight $w_{i,j,t}$ (the loss weighting side) and does not participate in the forward computation of $\\epsilon_\\theta$, the gradient of $\\mathcal{L}\_{\\text{gen}}$ with respect to $\\mathcal{M}^{pred}$ has one path, through $w_{i,j,t}$. Starting from the adaptive weight definition $w_{i,j,t} = (1 + \\alpha(k) \\cdot \\mathcal{M}^{pred}\_{i,j,t}) \\times (\\epsilon^{gen}\_{i,j,t} + \\delta)^\\gamma$, we have $\\frac{\\partial w_{i,j,t}}{\\partial \\mathcal{M}^{pred}\_{i,j,t}} = \\alpha(k) \\cdot (\\epsilon^{gen}\_{i,j,t} + \\delta)^\\gamma$. Applying the chain rule to $\\mathcal{L}\_{\\text{gen}} = \\mathbb{E}[\\sum w_{i,j,t} \\cdot l_{i,j,t}]$ where $l_{i,j,t} = \\lVert \\epsilon\_\\theta(\\mathbf{z}\_t, t)\_{i,j} - \\epsilon\_{i,j} \\rVert\_2^2$:
> $$\\frac{\\partial \\mathcal{L}\_{\\text{gen}}}{\\partial \\mathcal{M}^{pred}\_{i,j,t}} = \\alpha(k) \\cdot (\\epsilon^{gen}\_{i,j,t} + \\delta)^\\gamma \\cdot l_{i,j,t}$$
> The $\\mathcal{O}(\\epsilon^{gen})$ in the original manuscript arose from substituting the latent-space noise prediction error $l_{i,j,t}$ with the pixel-space reconstruction error $\\epsilon^{gen}$ for notational compactness.
>
> The reason why difficult regions receive higher weights: The core adaptive property follows directly from the closed-form definition of $w_{i,j,t}$. Since $\\gamma > 0$, the difficulty weighting term $(\\epsilon^{gen}\_{i,j,t} + \\delta)^\\gamma$ is monotonically increasing with respect to $\\epsilon^{gen}\_{i,j,t} = \\lVert \\hat{\\mathbf{x}}\_{i,j,t} - \\mathbf{x}^{clean}\_{i,j,t} \\rVert\_2^2$. Regions with poor current reconstruction quality have large $\\epsilon^{gen}$, thus receive larger $w$, and their contribution to the diffusion loss is amplified. This drives LoRA parameter updates to prioritize reducing noise prediction error in these regions, which can be regarded as a spatial extension of the focal loss principle where hard regions get upweighted.
>
> **Q3.** We appreciate this concern. The reviewer is right that thresholded pixel differences can be affected by lighting change, subtitle transparency, blur, and compression artifacts. In practice, we also apply an area-ratio filter since subtitles typically occupy 0.5%–2% of a frame. Under-detection (<0.5%) are excluded as zero-information frames. Over-detection (>2%) are typically caused by lighting or shadow changes being misidentified as subtitle regions and are removed. In our experiments, less than 1% of training frames are filtered. More importantly, Stage II is designed to handle this noise: the difficulty weighting term $(\\epsilon^{gen})^\\gamma$ in $w_{i,j,t}$ re-calibrates attention based on actual reconstruction error rather than blindly trusting the prior, so regions where noisy priors mislead the model receive corrective emphasis while falsely detected regions are naturally downweighted.
>
> **Q4.** The reviewer raises a valid and practically important concern. We offer two clarifications.
>
> First, the reported 4.86 s/frame corresponds to native 1280\\(\\times\\)720 generation, which directly contributes to the higher per-frame cost compared to methods operating at lower resolutions. To quantify this trade-off, we conducted a supplementary experiment using the CogVideoX-2B backbone within the CLEAR framework. The inference speed improves to **2.93 s/frame**, but the output is 480p and then upsampled, with visibly lower quality than our default setting.
>
> Second, CLEAR's architecture is naturally amenable to chunk-level parallelism for engineering deployment because CLEAR processes each chunk independently without requiring cross-chunk mask propagation. In practice, we segment long videos into temporal chunks and distribute them across multiple GPUs for parallel inference, with results concatenated. This yields near-linear speedup with the number of available GPUs.

---

> > ### Author Rebuttal · Reviewer_uFHP · 2026-04-02
> >
> > Thanks for the detailed reply from the author. The author's reply answered my question.

---

> > > ### Author Response · Authors · 2026-04-07
> > >
> > > We sincerely thank you for the valuable feedback and for taking the time to read our rebuttal carefully! We are very grateful that our response was able to address your concerns and clarify the questions you raised.

---

### Decision · Program_Chairs · 2026-04-30

**Decision:**

Accept (spotlight)

**Comment:**

This paper presents CLEAR, a two-stage mask-free end-to-end framework for video subtitle removal that addresses the critical practical limitation of existing methods, which universally require manual mask annotations during training and external text detection modules during inference. All four reviewers provided positive assessments of the work: two reviewers (MoSV and ax1m) assigned scores of 6 (Strong Accept), two reviewers assigned scores of 4 (Weak Accept). The authors submitted an exceptionally thorough, responsive, and well-documented rebuttal that comprehensively addressed every single concern raised by the reviewers, including providing complete mathematical derivations, detailed training and inference cost breakdowns, cross-backbone generalization experiments on CogVideoX, quantitative validation of prior accuracy, and clear discussions of limitations and future extensions. The work is widely recognized for its clear and impactful motivation, elegant modular design, exceptional parameter efficiency (training only 0.77% of the base diffusion model parameters), strong empirical performance that outperforms all state-of-the-art baselines, and remarkable zero-shot generalization across six unseen languages without language-specific fine-tuning. Most importantly, this paper fills a critical gap in practical video processing pipelines, as there is currently no publicly available end-to-end solution for this ubiquitous task, and CLEAR will have immediate and transformative value for a wide range of industrial applications including content localization, media editing, and large-scale video dataset cleaning. Based on the unanimous positive reviewer consensus, the complete resolution of all raised issues, and the exceptional practical significance of this work for the broader communities, I recommend Accept.